# The Impact of Income and Social Capital on the Health of People with Developmental Disabilities

**DOI:** 10.3390/healthcare10081543

**Published:** 2022-08-15

**Authors:** Bogcheon Choi, Eunsil Yi

**Affiliations:** 1Department of Rehabilitation, Jeonju University, Jeonju 55069, Korea; 2National Pension Research Institute, National Pension Service, Jeonju 54870, Korea

**Keywords:** people with developmental disabilities, self-rated health, social capital, income

## Abstract

This study examines the impact of income and social capital on the health of people with developmental disabilities, focusing on the moderating effects of income and social capital on health. Hierarchical regression analysis was conducted using data from 235 people with developmental disabilities who participated in the second wave of the Disability and Life Dynamics Panel. The findings show that people with developmental disabilities who were female, employed, and did not have multiple disabilities and chronic diseases were more likely to display higher levels of self-rated health. Furthermore, self-rated health was higher in those earning a higher income. The social network had a significantly positive effect on health, but its moderating effect on the impact of income did not carry statistical significance. Trust was found to have a moderating effect on the impact of income on health, where the group with greater trust and lower income was healthier than the group with lower trust. The findings suggest the need to provide income support and establish social capital for people with developmental disabilities to improve their health, and this study offers related policy implications.

## 1. Introduction

Health is at the very center of the life that everyone pursues. Maintaining a healthy state plays a key role for an individual to be active and productive in society [1], and health becomes an even more important resource to those with disabilities. Health and disability are closely interrelated. To people with disabilities, their impairments can cause diseases; conversely, chronic diseases and disorders may lead to disabilities. In fact, compared to those without disabilities, aging progresses at a faster rate in disabled people, and diseases such as polio, rheumatoid arthritis, and stroke occur more frequently among them [2]. Health problems in people with developmental disabilities are even more severe. According to the National Rehabilitation Center, the average age of death of people with disability is 76.1 years. However, the number is significantly lower for those with intellectual disabilities, at 55.5 years [3]. The disparity is more evident when compared to the average life expectancy of South Koreans, at 82.4 years. This situation is related to the low health levels of people with developmental disabilities. These people show higher chronic disease morbidity rates, including higher risk for obesity, cardiac problems, and high blood pressure, than those without disabilities. They also participate more in dysfunctional physical lifestyles, such as being less physically active, smoking, and consuming alcohol [2,4,5,6,7]. These behaviors lead to secondary conditions, cause limitations in everyday life [8], and may be the cause of higher suicide and death rates [7,9].

The health vulnerability of people with developmental disabilities is related to the characteristics of developmental disability—lower intellectual capacities, communication abilities, and social skills. Such characteristics make it challenging for an individual to self-recognize health problems and seek help from others through appropriate communication. Furthermore, it is difficult for others to spot symptoms of disease in people with developmental disabilities due to their limited activeness [6]. These characteristics of people with developmental disabilities obstruct their access to and usage of appropriate medical services, leading to them underestimating their health problems and further deterioration.

Most prior studies on the health of people with developmental disabilities have primarily focused on the health gap between people with and without disabilities or between people with developmental disabilities and those with other types of disabilities from the aspect of health inequality [2,6,10,11,12]. These studies shed light on the health inequality of people with developmental disabilities and have attracted interest, but there is a lack of research identifying the relationship between social factors and the health of people with developmental disabilities.

Generally speaking, genetic factors, health behavior and medical factors, and social factors are considered determinants of health. Social factors, in particular, have recently attracted attention for their impact on health. Interest in social factors as determinants of health have increased with a shift in how healthiness is defined—away from the traditional sense of being in a disease-free state to the newer perspective of environmental and social well-being [13,14]. Among the many social factors, income and social capital have a significant impact on health [15,16,17,18]. Higher income offers better access and use of health-related services, facilitating good health.

Social capital has been defined in a variety of ways by scholars. According to Bourdieu [19], social capital is actual or potential resources which are linked to the possession of a durable network of more or less institutionalized relationships of mutual acquaintance and recognition. Putnam viewed social capital as the characteristics of social organization such as trust, norms, and networks that can improve the efficiency of society by promoting harmonious behavior [20]. In other words, social capital is inherent in close relationships between individuals and may be considered characteristically equal to trust, norms, and networks. In general, social capital is a predictor of good health [21], and it is known to deter risky health behaviors such as smoking and alcohol consumption while improving usage of and accessibility to medical services. In particular, social relationships and trust function as a buffer against everyday stress, provide psychological comfort, and contribute to forming a healthy lifestyle, which all positively impact physical and mental health [20,21,22,23]. Furthermore, social capital serves as a vital resource for the physical and mental health of socially vulnerable groups such as older adults, married immigrants, people with disabilities, and rural community residents [21,24,25,26,27,28].

This study examines the impact of social factors, such as income and social capital, on the health of people with developmental disabilities. Developmental disabilities generally occur early in life and persist during one’s lifetime. This is why people with developmental disabilities experience prejudice and stigma throughout their life, limiting their participation in social activities and making it difficult to build social capital [12]. By examining the impact of these factors on health, this study seeks to determine social support measures that can ensure a healthy life for people with developmental disabilities. To this end, this study examines social networks and trust as essential elements of social capital for people with developmental disabilities. Although the indicators and methods for measuring social capital are as diverse as the definition of the concept [21], one factor, trust, is formed through networks between individuals [29] and thus can be viewed as a basic component of social capital. As social capital exists in relationships between people, social networks could be crucial for those with developmental disabilities. Moreover, they may indicate their trust in other people and communities depending on how they perceive social attitudes directed toward them. Furthermore, this study tests the moderating effect of social capital whether it can ease the effects of poverty on health since social capital is known to act as a buffer between health and everyday life stress.

## 2. Materials and Methods

### 2.1. Research Participants and Data

This study utilized data from the Disability and Life Dynamics Panel (DLDP) conducted by the Korea Disabled People’s Development Institute. The panel survey on the lives of people with disabilities identifies the patterns and causes of changes experienced by such individuals and their families. It is a nationwide survey that was developed to prepare basic data required for the establishment of welfare policies for those with disabilities in Korea [27]. The survey population is people with disabilities living in Korea, and it investigates disability acceptance and change, health and medical care, self-reliance, and social participation; thus, the variables related to social capital, income, and health necessary for this study needed to be constructed. This study used data from the second year of the survey, 2019. Altogether, 5527 people with disabilities in Korea responded, of which 367 people (6.64%) had developmental disabilities. This study analyzed 235 adults with developmental disabilities, excluding children with disabilities under the age of 19.

### 2.2. Variable Measures

#### 2.2.1. Outcome Variable

Self-rated health of developmentally disabled people was considered as the outcome variable of this study. Subjective evaluation of one’s own state of health is a helpful indicator of the overall state of health, including physical health, disease, and disability. It is known to predict the actual health state of an individual with fair accuracy [21,28]. Self-rated health in this study was measured using a 4-point Likert scale, where a higher score indicates a better state of health.

#### 2.2.2. Predictor Variables

Income and social capital are the predictor variables in this study. Income was measured as the average monthly income of a household, and the natural log for the value was used. Social capital consisted of social networks and trust. The social network was measured as the degree of support a participant received from family members and other people, using the questions “Do you receive emotional help and support from family members?” and “Do you receive emotional help and support from other people?”. The average score between the two questions based on a 4-point Likert scale was applied as a measure for social networks. A higher score shows that an individual’s social network is well-formed. Trust generally implies trust in other people and the social community. Trust in the context of this study is measured as the social awareness of disabled people perceived by them since a higher level of trust can lead to holding a more positive view of society. A 4-point Likert scale was used to evaluate responses to the following statements: “I believe I am separated from society”, “I do not find the society’s view on disabilities to be appropriate”, “I prefer not to take part in external activities because of my disability” and “I try not to express my emotions because of my disability” The scores were reverse coded so that higher scores signify a greater level of trust, and the average score between the four statements was used.

#### 2.2.3. Covariates

Age (0 = less 50, 1 = 50 or more), sex (0 = female, 1 = male), level of education (0 = less than high school), employment status (0 = unemployed, 1 = employed), presence of multiple disabilities (0 = yes, 1 = no), presence of chronic disease (0 = yes, 1 = no), and the number of household members. All factors which could impact health were controlled.

### 2.3. Analysis/Methods

This study first conducted descriptive and frequency analyses to identify the general characteristics of the research participants and the characteristics of the major variables. Then, the correlations between the major variables were verified. Finally, a hierarchical regression analysis was conducted to examine the impact of developmentally disabled people’s income and social capital on self-rated health. Mean-centering was applied to income and social capital variables to analyze the moderating effects and address the issue of multicollinearity. SPSS 25.0 was employed for statistical analysis.

## 3. Results

### 3.1. General Characteristics

Table 1 summarizes the general characteristics of the research participants. There were more male participants (53.6%) than female participants (46.4%). The age group 19–26 was the largest with 41.3%, those in their 30s accounted for 14.9%, those in their 40s accounted for 15.3%, those in their 50s made up 13.2%, and those 60 and above made up 15.3% of all participants. Participants with high school education and higher were most numerous (63.4%), followed by middle school education (16.2%), primary school education (12.3%), and no education (8.1%). Further, 8.5% of participants had multiple disabilities, while 91.5% did not. More participants had chronic diseases lasting over 3 months (65.4%) than those who did not suffer from chronic diseases (34.6%). Regarding the number of household members, 13.6% lived alone, 26.4% lived in a two-person household, 31.1% lived in a three-person household, and 28.9% lived in a household with four or more people, and 74.9% of participants were unemployed, while 25.1% went to work.

### 3.2. Characteristics of Major Variables and Their Correlations

Table 2 shows the characteristics of the major variables. The average health score for developmentally disabled people was 2.66 (SD = 0.63), and their average monthly income was KRW 2,487,300, for which the natural log was 5.24 (SD = 0.77). Regarding the social capital level, the average social network score was 2.97 (SD = 0.60), and the average trust score was 2.42 (SD = 0.52), a value slightly lower than that for the social network. The skewness of the major variables did not surpass the absolute value of 3, and the kurtosis was lower than the absolute value of 7, satisfying the assumptions of normal distribution.

Table 3 shows the correlations between the major variables. There were significant correlations between income and social network (r = 0.191, *p* < 0.01) and income and health (r = 0.274, *p* < 0.01). Additionally, there were significant correlations between social network, the variables of social capital, and trust (r = 0.142, *p* < 0.05) and between social network and health (r = 0.308, *p* < 0.05).

### 3.3. Hierarchical Regression Analysis Results

Table 4 shows a hierarchical regression analysis conducted to explore the impact of income and social capital on the health of people with a developmental disability. Demographic covariates were included in stage 1, and the predictor variables of income and social capital (e.g., social network and trust) were applied in stage 2. Then, the interactive term between income and social capital was included in stage 3 to observe whether social capital has a moderating effect on the influence of income on health.

Demographic factors in model 1 explain 19.9% of developmentally disabled people’s self-rated health. It was found that the score for state of health was higher for disabled females (β = −0.310, *p* = 0.003), those with more household members (β = 0.091, *p* = 0.016), employed disabled people (β = 0.273, *p* = 0.002), those without multiple disabilities (β = 0.319, *p* = 0.018), and those without chronic diseases (β = 0.235, *p* = 0.006).

In model 2, the social factors of income and social capital that affect health are added to model 1. Social capital consists of social networks and trust. According to the analysis, model 2 explains 26.6% of the self-rated health of research participants. With respect to demographic factors, sex, employment status, the presence of multiple disabilities, and the presence of chronic diseases displayed statistical significance. A higher income (β = 0.084) suggested better self-rated health, showing statistical significance at the *p* < 0.10 level. The variable of a network of social capital was found to have a significant effect on self-rated health (β = 0.264, *p* = 0.000), but the same cannot be said for the impact of trust.

The interactive term between income and social capital was input in model 3. The sex, employment status, presence of multiple disabilities, and presence of chronic diseases of people with developmental disabilities were found to have statistical significance on self-rated health scores. Furthermore, income (β = 0.134, *p* = 0.014) and social network (β = 0.264, *p* = 0.000) had a statistically significant and positive effect on their health, suggesting that developmentally disabled people with higher incomes and better developed networks were healthier. Regarding interactive terms, that between income and trust showed statistical significance, demonstrating a moderating effect. Although social network did have a statistically significant influence on health, a significant moderating effect was not present in the relationship between income and health. Trust, however, did not significantly impact health but was found to show a moderating effect on the influence of income over health.

Figure 1 shows the moderating effect of trust. Developmentally disabled people with high trust levels demonstrated better health even if they had low income, and maintained consistency in their self-rated health when their income increased. However, in the case of low income, those with low trust levels showed a worse state of health than those with high trust levels. Their self-rated health tended to improve dramatically with income increase. In other words, developmentally disabled people with high trust levels had high self-rated health regardless of their income size, but those with low trust levels had low-self rated health, which improved substantially with an increase in income. This finding suggests that the impact of income on self-rated health depends on the level of social capital, where those with lower trust levels react more sensitively to the influence of income.

## 4. Discussion

This study sought to identify the impact of income and social capital on the health of people with developmental disabilities with the aim of verifying the influence of social factors that determine their health. To this end, an analysis was conducted of intellectually disabled people participating in the second wave of the Disability and Life Dynamics Panel.

According to the findings, developmentally disabled people with higher household income and better-developed networks scored higher in self-rated health. Moreover, self-rated health was better in disabled females, employed people with developmental disabilities, and those who did not have multiple disabilities and chronic diseases. With respect to the moderating effect of income and social capital, trust was found to have a moderating effect on the influence of income over self-rated health. These findings have the following implications:

First, self-rated health was better for developmentally disabled people with higher household income, which is consistent with the findings of prior studies [4,23,24] that determined that poverty or socio-economic status has a significant effect on health. This outcome implies the need to elevate the income levels of people with developmental disability so they can lead healthy lives. According to the 2020 National Survey on Persons with Disabilities, 36.3% of intellectually disabled people’s households were beneficiaries of livelihood benefits provided through the National Basic Living Security System [29]. This share is relatively large compared to that of all disabled people’s households at 19.1%, indicating the impoverished life of intellectually disabled people. Furthermore, the difference is apparent in the income level of disabled individuals, where the average monthly income of intellectually disabled people (KRW 573,000 per month) is much less than that of all disabled people (KRW 1,003,000 per month). Evidently, the low income earned by people with developmental disabilities acts as a factor that deteriorates their health. Hence, policy measures should be established to raise the income level of these people. Oftentimes, developmentally disabled people fail to find employment, and when they do, they are given low-ranking positions in the job market that offer little job stability and pay very low wages. Due to such circumstances, these people do not receive sufficient income through work, becoming more reliant on public transfer income. It is therefore necessary to support the opportunity for developmentally disabled people to earn income by participating in the job market. Policies are needed to expand jobs that match the characteristics and needs of developmentally disabled people while also helping them maintain their jobs security. In addition, a policy that provides income support for the households of people with developmental disabilities should be considered. Currently, cash benefits for disabled people, such as the disability pension and the disability (child) allowance, largely apply to only the individual with a disability. However, the reality is that these benefits are insufficient and they cannot even sufficiently cover additional expenses that occur due to disability. To this end, in addition to strengthening income support for disabled people, an allowance for their families should be provided to guarantee the income of their household. It is also possible to support their household income by offering indirect support that reduces expenses that occur as a result of disability.

Second, the impact of social capital on self-rated health exhibited varying outcomes depending on the sub-factors of social capital. The social capital sub-factor that influenced self-rated health was social network—developmentally disabled people with better-developed networks displayed higher self-rated health levels. The findings support prior studies [24,30,31,32,33,34] that demonstrated the impact of social networks on health. Trust, however, did not have a significant direct impact on health in this study. This was different from the results by Purtinga (2006), who examined data from 22 European countries and reported a strong association between social trust and subjective health [35].

The significant correlation between social network and health indicates that developmentally disabled people who are strongly connected to family and society can receive adequate support from those relationships, leading them to live a healthy life. It implies the need to reinforce support for people with developmental disabilities so that they can expand their social networks to build a healthy life. While forming diverse relationships is important in their network, the more vital issue lies in the interaction of the relationships within the network and receiving social support. According to a survey by the Korea Disabled People’s Development Institute, 53.3% of developmentally disabled people said they did not have any close friends, neighbors, or acquaintances in their life, while 31.3% said they had fewer than two people with whom they are close [27], exposing the extremely shallow and limited social network of these people. Nonetheless, they receive emotional help and support from informal networks, including family and others around them, and the degree of help and support from social welfare facilities or organizations is quite weak. Hence, for people with a developmental disability to build social capital, it is necessary to both expand social participation and form networks and to provide proper help and support through them.

Third, upon examining the moderating effect of income and social capital on health, trust level, a component of social capital, was found to demonstrate it. Individuals with a high level of trust exhibited better health even if they earned a low income. Although trust did not have a direct impact on health, it was able to positively affect the health of the low-income group. Such a finding is consistent with the analysis of Wahl et al. [17], which examined the correlation between social capital and health for recipients of public aid in Norway. It demonstrated that even as recipients of public aid, those with a high level of social capital showed significantly better mental health and quality of life [17]. Furthermore, the abovementioned finding is also partially consistent with Yu et al. [22], which confirmed that the self-rated health of disabled people improved with a higher level of trust and with the research that verified the positive impact of social trust on the health of the elderly [24,26]. In other words, social capital built upon trusting relationships positively influences the health of not only physically disabled people and older adults but also that of developmentally disabled people. This implies the need to raise the trust level of people with developmental disabilities by building an environment that ensures their safety and health within local communities.

As Bourdieu stresses, trust is the social capital that is formed in relationships [36]. In order to form relational trust between people with developmental disabilities and people without disabilities, a positive perception of community members with developmental disabilities is of utmost importance, as is expanding opportunities for social participation that can improve the general population’s understanding of such individuals. Improved friendly attitudes toward developmentally disabled people and their social inclusion increase their level of trust, which in turn, positively affects their health. Hence, much effort should be made to realize the social inclusion of developmentally disabled people by creating unbiased and safe local communities to raise these people’s trust in society. Moreover, 52.3% of intellectually disabled people feel discriminated against because of their disability, and 30.8% have experienced abuse and violence [29]. The large share of developmentally disabled people experiencing discrimination and violence exposes our society’s overall low awareness of their rights. It is therefore necessary to provide support to improve and raise awareness regarding the rights of people with developmental disabilities, and actively carry out interventions when abuse and violence occur. Such actions will offer them more opportunities to experience a warm welcome and kindness from their local communities, nurturing a sense of trust that they are protected in a safe environment.

There has been much research on the health disparity of developmentally disabled people globally, but not enough studies on this topic have been conducted in Korea. Given such a circumstance, this study is meaningful in that it empirically verified the impact of social factors on the health of developmentally disabled people. By confirming determinants of health to reduce health inequality of these people, this study offers the basic data needed to establish practical and policy strategies to promote health. Nevertheless, this study has limitations accompanied by using secondary data. Although it measured the quality of the social network, a component of social capital, it did not measure the size and frequency of the network due to limitations in variables. Furthermore, the research participants were limited to intellectually disabled people, leading to the limitation of generalizing the results of those with autistic disorders and other types of disabilities. Further, because this study used cross-sectional data, an additional limitation is that it did not examine the processes and changes among which social factors affect health. In spite of the limitations, this study is expected to contribute to establishing measures to reduce health disparity among developmentally disabled people and supporting their good health in everyday life.

## 5. Conclusions

Good health is important to everyone [37], including people with developmental disabilities, one of the most vulnerable groups in society. This study analyzed the impact of developmentally disabled people’s income and social capital on their health. Results from 235 research participants with developmental disabilities showed income and social capital influenced their health, and the moderating effect of social capital was confirmed. This study also verified that health level was lower for disabled men, unemployed disabled people, those with multiple disabilities, and those with chronic diseases. Furthermore, the self-rated health of people with a developmental disability was better with higher household income and the establishment of a social network and support system. This suggests the need to support the building of social capital, including social relationships and trust, in addition to income support to promote the health of people with developmental disabilities. Through this, the right for people with developmental disabilities to improve their well-being and maintain a healthy life should be guaranteed, as should the opportunity to access the resources necessary to live a good life.

## Figures and Tables

**Figure 1 healthcare-10-01543-f001:**
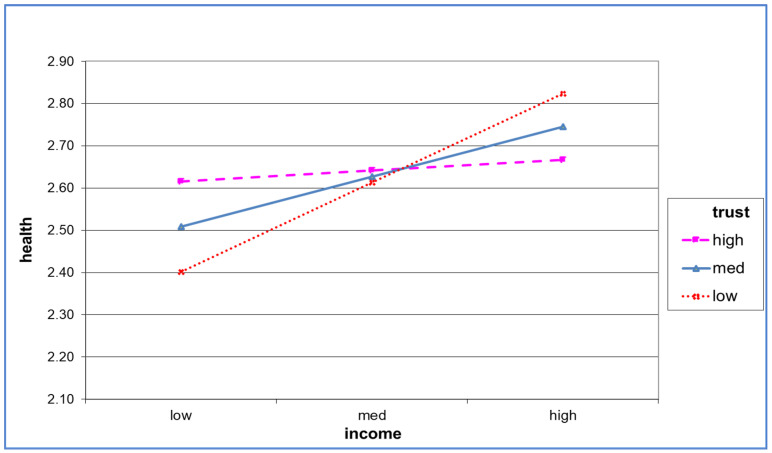
The moderating effect of trust.

**Table 1 healthcare-10-01543-t001:** General characteristics of the research participants.

Variables		Frequency (Number of People)	Ratio (%)
Sex	Female	109	46.4
	Male	126	53.6
Age	19–29	97	41.3
	30–39	35	14.9
	40–49	36	15.3
	50–59	31	13.2
	≥60	36	15.3
Level of education	No education	19	8.1
	Primary school	29	12.3
	Middle school	38	16.2
	High school and above	149	63.4
Presence of multiple disabilities	Yes	20	8.5
	No	214	91.5
Presence of chronic diseases	Yes	153	65.4
	No	81	34.6
Number of household members	1	32	13.6
	2	62	26.4
	3	73	31.1
	4 or more	68	28.9
Employment status	Unemployed	176	74.9
	Employed	59	25.1

**Table 2 healthcare-10-01543-t002:** Characteristics of the major variables.

Variables	Mean	SD	Min.	Max.	Skewness	Kurtosis
Health	2.66	0.63	1	4	−0.518	0.295
Income (ln)	5.24	0.77	3.04	7.79	0.005	0.522
Network	2.97	0.60	1	4	−0.453	0.797
Trust	2.42	0.52	1	4	0.048	0.058

**Table 3 healthcare-10-01543-t003:** Correlation between the major variables.

Variables	Income (log)	Network	Trust	Health
Income (log)	1			
Network	0.191 **	1		
Trust	−0.023	0.142 *	1	
Health	0.274 **	0.308 **	0.074	1

* *p* < 0.05, ** *p* < 0.01.

**Table 4 healthcare-10-01543-t004:** Hierarchical regression analysis and moderating effect analysis results.

	Model 1				Model 2				Model 3		
	B	95% CI		*p*-Value	B	95% CI		*p*-Value	B	95% CI	*p*-Value
Constant	2.361	2.063, 2.659		0.000	2.588	2.265, 2.911		0.000	2.627	2.305, 2.948	0.000
Age (ref: <50)	−0.039	−0.187, 0.109		0.605	−0.038	−0.180, 0.104		0.598	−0.023	−0.166, 0.120	0.750
Sex (ref: female)	−0.310	−0.517, −0.104		0.003	−0.308	−0.507, −0.109		0.003	−0.320	−0.517, −0.122	0.002
Level of education (ref: <high school)	−0.057	−0.245, 0.131		0.551	−0.127	−0.310, 0.056		0.172	−0.144	−0.326, 0.037	0.119
Number of household members (in people)	0.091	0.017, 0.164		0.016	0.035	−0.049, 0.120		0.409	0.022	−0.062, 0.106	0.601
Employment status (ref: unemployed)	0.273	0.099, 0.447		0.002	0.212	0.039, 0.385		0.016	0.222	0.051, 0.393	0.011
Multiple disabilities (ref: yes)	0.319	0.583, 0.056		0.018	0.374	0.631, 0.117		0.005	0.385	0.640, 0.130	0.003
Chronic disease (ref: yes)	0.235	0.069, 0.400		0.006	0.218	0.059, 0.378		0.008	0.216	0.058, 0.374	0.008
Income					0.084	−0.016, 0.184		0.099	0.134	0.027, 0.240	0.014
Network					0.264	0.142, 0.387		0.000	0.264	0.143, 0.385	0.000
Trust					0.042	−0.084, 0.167		0.514	0.024	−0.102, 0.149	0.709
Income × network									0.020	−0.128, 0.168	0.791
Income × trust									−0.177 *	−0.316, −0.039	0.012
F	9.279 ***				9.476 ***				8.697 ***		
adj. R-squared	0.199				0.266				0.283		

* *p* < 0.05, *** *p* < 0.001.

## Data Availability

The data used in this study can be obtained from the Korea Disabled People’s Development Institute. http://www.koddi.or.kr (accessed on 11 July 2022).

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
