# Peer review of "The Impact of Income and Social Capital on the Health of People with Developmental Disabilities"

_healthcare, 2022, doi:10.3390/healthcare10081543_

Round 1
Reviewer 1 Report
The topic addressed by the authors of the manuscript is quite relevant, and the approach is clear and objective. However, it is necessary to work a little more on the foundation of the study, the state of the art (In the introduction section).
In the Materials and Methods section, the authors should clarify or discuss the data collection a little more, and indicate the repository (access considering that they mention having used data in the public domain).
In the results section, it would have been interesting to include some current data in order to establish some comparatives that allow identifying changes (improvement or worsening) in the aforementioned analyses.
Improve the graph in Figure 1.
The manuscript has potential, and the approach is interesting. However, further substantiation and insertion of comparison data are required.
Author Response
"Please see the attachment"

Reviewer 2 Report
The authors do interesting and quality work. In this sense, they deal with the study of Korea, an area that has not been previously analysed. The research turns out to be a good contribution to the scientific community. However, in general, I believe that the results need to be compared with those of other regions or with previous papers. In addition, it would be convenient to expand the methodological information, as well as the discussion and conclusion. The intention of these comments is to help authors to obtain a paper with greater international impact. In this sense, please do not understand that I intend to make a negative criticism of their work. In fact, I believe that the work has quality and is a good contribution.
In line 57, when the authors said “but there is a lack of research identifying factors that determine the health...”, it seems that reference is made to social factors, but I think it would be appropriate to specify this. Since we are talking about policies, I think it would be somewhat clearer and would introduce the reader to the subsequent ideas.
In the methodological section, it would be convenient to indicate expressly that it has been carried out in Korea. It would also be important to explain in depth what the Disability and Life Dynamics Panel (DLDP) consists of. How has the information been obtained? Are the data for the 235 adults all the data that were available? Have some people been excluded? If so, what were the exclusion criteria? In what geographic area did the informants live? I believe that all this information could help to better understand the sample used. In the case that you do not have such data, it would be convenient to justify it. It is possible that if the authors could provide an explanatory table with this information, it would be sufficient.
I find the main limitation of this paper in the discussion. It would be convenient to expand this section. I believe that an effort should be made to contrast the data with other regions, with other disabilities, etc. In addition, it would also be really interesting if the authors could rely more on Bourdieu's work to complete this section. Regarding the discussion, I believe that it would be important to make a comparison with other geographical areas, trying to compare, as far as possible, countries. This would give a greater internationalization value to the paper.
It could be useful this: Stronks, Karien, Toebes, Brigit, Hendriks, Aart, Ikram, Umar & Venkatapuram, Sridhar. (2018). Social justice and human rights as a framework for addressing social determinants of health: final report of the task group on equity, equality and human rights. Review of social determinants of health and the health divide in the WHO European Region. World Health Organization. Regional Office for Europe. https://apps.who.int/iris/handle/10665/350401
Regarding social capital in discussion, I humbly consider that further discussion is needed. As indicated in this review (Ehsan et al, 2019: https://doi.org/10.1016/j.ssmph.2019.100425) many aspects of the study of social capital remain unclear. For this reason, I suggest discussing these aspects in this paper and show what this research contributes to the academic debate.
The authors focus their interest on personal trust. However, we know that trust is a fundamentally interpersonal process. In this sense, I suggest that this aspect should be explored in greater detail to explain this phenomenon in greater detail. In fact, at the beginning of the paper the authors mention Bourdieu but his work is hardly used in this research. As an orientation, I suggest reading this interesting work that I believe could help to improve the discussion: Morten Frederiksen (2014) Relational trust: Outline of a Bourdieusian theory of interpersonal trust, Journal of Trust Research, 4:2, 167-192, DOI: 10.1080/21515581.2014.966829
On the other hand, although it is not one of the aspects under study, I believe that mention could be made of overprotection as an element related to social capital. The following paper (Buell & Chadwick, 2017) makes some mention of this and I believe, once again, that it would allow the research presented to have greater depth and quality: https://doi.org/10.1177/1744629517707086
Finally, I suggest that you expand on the conclusions a bit more by reviewing the highlights of the paper.
Author Response
"Please see the attachment"

Round 2
Reviewer 1 Report
Dear Authors
I believe that the observations made on the manuscript were duly attended to and justified.
I would just like to know how the development of low-cost assistive devices for people with disabilities could help or improve access to healthcare https://www.mdpi.com/2411-9660/5/4/75. It is possible that the question is a little outside the objectives of the manuscript, however, it is also possible that it will be worth the analysis, even if in a future way.
Regards
Author Response
Dear Reviewr,
Thank you for your thoughtful suggestions and insights regarding assistive devices.
However, we will appreciate your understanding that it is difficult to reflect your comments in the manuscript for the following reasons. Since the purpose of this study is to examine the impact of income and social capital on health, your points regarding assistive devices are beyond the scope of this paper. Please also consider that the participants of this study are people with intellectual disabilities. Because they do not use normally assistive devices, it can be said that there is little effect of assistive devices on their health. However, for people with physical disabilities, the policy implications of assistive devices in maintaining their health will be important.
Thus, even though we could not reflect your comment in the manuscript, it will provide important implications for our follow-up studies examining the relationship between characteristics of disability and health. Although major modifications have not been made to the content, we tried to make some expressive corrections in order to improve completeness and accuracy in the final stage.
Thank you for your consideration.
Reviewer 2 Report
The authors have made the minimum changes so that it can be accepted. Now the text has a little more continuity and the methodology has been somewhat more detailed. The changes to the conclusion are very limited. In any case, I believe that the article can be published in the current version. However, I am sure that if the changes had been more profound the paper could be fantastic.
Author Response
We thank you for your kind comments and positive evaluation of this study. Although major modifications have not been made to the content, we have tried to make some expressive corrections in order to improve completeness and accuracy in the final stage.
Thank you for your consideration.